# Dual-Energy CT Iodine Uptake of Head and Neck: Definition of Reference Values in a Big Data Cohort

**DOI:** 10.3390/diagnostics14050496

**Published:** 2024-02-26

**Authors:** Ibrahim Yel, Giuseppe Mauro Bucolo, Scherwin Mahmoudi, Vitali Koch, Aynur Gökduman, Tommaso D′Angelo, Leon David Grünewald, Mirela Dimitrova, Katrin Eichler, Thomas J. Vogl, Christian Booz

**Affiliations:** 1Goethe University Frankfurt, University Hospital Frankfurt, Clinic for Radiology and Nuclear Medicine, 60590 Frankfurt, Germany; 2Department of Biomedical Sciences and Morphological and Functional Imaging, University of Messina, 98122 Messina, Italy

**Keywords:** iodine contrast media, iodine quantification, iodine uptake reference values, computed tomography, head and neck imaging

## Abstract

Background: Despite a considerable amount of literature on dual-energy CT (DECT) iodine uptake of the head and neck, the physiologic iodine uptake of this region has not been defined yet. This study aims to establish reference values for the iodine uptake of healthy organs to facilitate clinical application. Methods: Consecutive venous DECT scans of the head and neck were reviewed, and unremarkable exams were included (*n* = 617). A total of 35 region of interest measurements were performed in 16 anatomical regions. Iodine uptake was compared among different organs/tissues and subgroup analysis was performed (male (*n* = 403) vs. female (*n* = 214); young (*n* = 207) vs. middle-aged (*n* = 206) vs. old (*n* = 204); and normal weight (*n* = 314) vs. overweight (*n* = 196) vs. obese (*n* = 107)). Results: Overall mean iodine uptake values ranged between 0.5 and 9.4 mg/mL. Women showed higher iodine concentrations in the cervical vessels and higher uptake for the parotid gland, masseter muscle, submandibular glands, sublingual glands, palatine tonsils, tongue body, thyroid gland, and the sternocleidomastoid muscle than men (*p* ≤ 0.04). With increasing age, intravascular iodine concentrations increased as well as iodine uptake for cerebellum and thyroid gland, while values for the tongue and palatine tonsils were lower compared to younger subjects (*p* ≤ 0.03). Iodine concentrations for parotid glands and sternocleidomastoid muscles decreased with a higher BMI (*p* ≤ 0.004), while normal-weighted patients showed higher iodine values inside the jugular veins, other cervical glands, and tonsils versus patients with a higher BMI (*p* ≤ 0.04). Conclusion: physiologic iodine uptake values of cervical organs and tissues show gender-, age-, and BMI-related differences, which should be considered in the clinical routine of head and neck DECT.

## 1. Introduction

CT imaging of the head and neck region has a central role in the diagnosis of acute and non-acute inflammatory processes, trauma, and malignancies. Due to its widespread availability and low costs compared to MRI, contrast-enhanced CT is frequently the first imaging method of choice used to obtain the necessary information in this complex anatomical region, despite its inferiority in soft tissue visualization.

However, following technological advances in dual-energy CT (DECT), CT-based diagnostics can be significantly enhanced and refined [1,2]. Courtesy of the dual-energy technology and the resulting sophisticated material decomposition analysis, virtual maps of the iodine uptake of tissues can be reconstructed and absolute values (in mg/mL) can be obtained, presenting a highly accurate, novel quantitative parameter [3,4,5,6,7,8,9].

Despite the extensive interest in utilizing iodine values as a measurable biomarker in abdominal CT—mainly to differentiate between benign and malignant lesions or quantitative perfusion analyses—there is hardly any application in the field of head and neck imaging. This shortfall is largely due to the constrained experience with DECT in this small yet anatomically complex region, compounded by the absence of established reference values for iodine uptake.

Recognizing the potential of DECT to enhance diagnostic imaging by offering molecular insights beyond traditional anatomical assessments, this study aims to define and establish reference values for the iodine uptake of healthy head and neck organs and significant anatomical landmarks. The capability to measure absolute iodine concentrations introduces a new dimension to imaging, enabling the accurate differentiation between pathological and non-pathological states, which is crucial for effective patient management in the head and neck region.

The development of a normative database for iodine uptake values is a critical step toward unlocking DECT’s clinical potential in the head and neck region, enhancing clinicians’ ability to interpret DECT images with greater confidence and precision. Such standardization paves the way for the integration of iodine concentration maps into routine diagnostic workflows and facilitates personalized treatment approaches.

Furthermore, quantifying iodine uptake through DECT could significantly advance the early detection of diseases in the head and neck area by providing objective criteria for assessing tissue characteristics. This early detection capability highlights DECT’s role not only in diagnosis but also in the proactive management of health conditions, offering a tool that improves conventional imaging modalities by identifying deviations from normal that may indicate the onset of disease.

In conclusion, by addressing the gap in DECT application within head and neck imaging and establishing reference iodine uptake values, this study lays the groundwork for future research aimed at expanding the clinical utility of this innovative imaging modality. Ultimately, the goal is to enhance the diagnostic accuracy of DECT, contributing to the advancement of personalized medicine and improving patient outcomes. 

## 2. Materials and Methods

### 2.1. Study Population

After approval of our monocentric, retrospective study by our institutional review board, we performed an analysis of our database for patients who had undergone clinically indicated, contrast-enhanced DECT scans of the head and neck region between December 2016 and December 2020, with no pathologies detectable in CT imaging (scans were performed in clinical routine, e.g., after trauma, to exclude infections or tumors, or to exclude cervical lymphadenopathies and other suspected cervical pathologies). The absence of pathologies was carefully confirmed by two independent radiologists before inclusion. Common laboratory values of all patients were within the normal range. None of the subjects had a history of known head/neck surgery, malignancy, or other known diseases. In a follow-up period of one year, none of the patients showed cervical symptomatology (confirmed via anamnestic interviews, clinical examinations, ultrasound examinations, or cross-sectional imaging). Further exclusion criteria were patients younger than 18 years, imaging artifacts, and patients with systemic treatment. Figure 1 depicts the flowchart of patient inclusion.

### 2.2. DECT Imaging Technique

All examinations were performed on the same third-generation dual-energy CT scanner (SOMATOM Force, Siemens Healthcare, Forchheim, Germany). Patients were examined in the supine position and CT images were acquired from the orbital roofs to the aortic arch. The study protocol consisted of a single venous-phase acquisition in DECT mode, which automatically started 70 s after the beginning of the injection of Iomeprol (Imeron 350, Bracco Imaging, Konstanz/Germany). Contrast media (1.2 mL/kg body weight) was injected into a superficial vein of the forearm at a flow rate of 3 mL/s. The following parameters were used for DECT imaging: 90 kV and 190 mAs per rotation on tube A; Sn 150 kV with tin filter and 95 mAs per rotation on tube B; and 0.5 s rotation time, 0.6 pitch, and 2 × 192 × 0.6 mm collimation. All image series were reconstructed using a 3.0 mm slice thickness in a 2.0 mm increment by using a soft-tissue (Bv40) kernel.

### 2.3. Iodine Mapping and Uptake Measurements

Dual-energy analysis was performed by using the commercially available post-processing software liver virtual non-contrast (VNC) (syngo.via VA30, Siemens Healthineers, Forchheim/Germany). Initially designed for iodine measurements in the liver, the liver VNC software has already been used for measurements in various anatomic regions [4,5,10,11]. The basis of its algorithm is a three-material decomposition of iodine, fat, and tissue. The liver VNC application enables the visualization of iodine (contrast agent) concentration by decomposing the iodine content out of the Hounsfield unit value of any voxel and displaying the pure iodine map as a colored overlay on the gray-scale image. The iodine slope was calculated automatically using the software at 90 keV and 150 keV. Settings were left on default (Resolution: 2, Maximum [HU]: 3071, Iodine Ratio 3.46).

By performing circular dual-energy region of interest (ROI) measurements on the generated maps, iodine density can be calculated (in mg/mL). The measurements were independently performed by two radiologists (with 6 and 15 years of experience, respectively) in all patients and mean values were calculated.

Depending on the size of the organ/region, ROI measurements were targeted to be between 0.4 cm^2^ and 1.0 cm^2^ and were carefully placed to avoid the inclusion of surrounding tissue. In total, 35 ROIs were placed in each study by 1 investigator and the corresponding absolute iodine concentration values for each ROI were extracted. To minimize the influence of patient-specific perfusions on the results, additional bilateral measurements of the intraluminal iodine concentration of the common carotid artery were performed to achieve data normalization by calculating the iodine ratio (absolute iodine concentration of tissue/iodine concentration in the common carotid artery). To verify the correctness of the measurements, an additional ROI was placed in an area where iodine uptake was not expected (inside the trachea). A detailed list of the distribution of the measurements can be found in Table 1. Examples of ROI placements are shown in Figure 2.

### 2.4. Statistical Analysis

Mean values were calculated for each organ and structure. For the overall analysis and the subgroup analyses, mean values and standard deviations (SD) were calculated. Numerical values of continuous variables were listed as mean values ± standard deviation. Gaussian data distribution was assessed using the Kolmogorov–Smirnov test. An unpaired *t*-test and analysis of variance (ANOVA) with Tukey multiple comparison post hoc tests were performed for normally distributed data. A Mann–Whitney U test and Kruskal–Wallis tests were applied in case of non-normal distribution. A statistically significant difference was defined by a *p*-value less than 0.05. Statistical analysis was performed by using GraphPad Prism Version 7.0 (GraphPad Software; La Jolla, CA, USA) and IBM SPSS Statistics Version 28 (IBM SPSS statistics; Armonk, NY, USA).

## 3. Results

### 3.1. Patient Collective

The final study cohort consisted of 617 patients (Caucasian: 596, Asian: 13, Black: 4, Hispanic: 4; mean age, 55.0 ± 18.1 years; range, 18–98 years), including 214 women (mean age, 54.4 ± 18.6 years; range, 18–98 years) and 403 men (mean age, 55.3 ± 17.8 years; range, 18–91 years).

Subgroups were defined regarding sex (male (*n* = 403) vs. female (*n* = 214)), age (18–48 (*n* = 207) vs. 49–64 (*n* = 206) vs. 65–98 (*n* = 204) years), and BMI (24.9 or below (*n* = 314) vs. 25–29.9 (*n* = 196) vs. 30.0 and above (*n* = 107) kg/m^2^).

### 3.2. Overall Iodine Values

There were no side differences in the same subjects for paired structures and organs regarding absolute iodine values (*p* > 0.43). Therefore, mean values were calculated for paired organs/structures.

Besides the contrast-media-enhanced jugular vein with 6.66 ± 2.7 mg/mL and the carotid artery with 5.88 ± 2.0 mg/mL, the highest absolute iodine concentration values were documented for the thyroid gland with mean values of 4.68 ± 1.5 mg/mL. The temporal lobes with mean values of 0.82 ± 0.4 mg/mL and the neck muscles (sternocleidomastoid: 0.76 ± 0.3 mg/mL; masseter: 0.99 ± 0.4 mg/mL) demonstrated low iodine uptake. The same trend was seen in the normalized iodine ratios. No iodine uptake was recorded in the control measurements inside the trachea (air).

A detailed overview of the overall mean absolute iodine values and iodine ratios is demonstrated in Table 2 and illustrated in Figure 3.

### 3.3. Impact of Sex

Women showed higher iodine concentrations in the carotid artery (6.31 ± 2.1 vs. 5.66 ± 1.8 mg/mL, *p* < 0.001) and in the jugular vein (7.56 ± 3.1 vs. 6.18 ± 2.3 mg/mL, *p* < 0.001) as well as higher iodine uptake for all cervical glands: the parotid glands (2.00 ± 0.8 vs. 1.76 ± 0.7 mg/mL, *p* < 0.001), submandibular glands (2.57 ± 1.3 vs. 2.27 ± 1.2 mg/mL, *p* = 0.004), sublingual glands (3.53 ± 1.0 vs. 3.35 ± 0.9 mg/mL, *p* = 0.04), and the thyroid glands (5.07 ± 1.6 vs. 4.48 ± 1.4 mg/mL, *p* < 0.001). Furthermore, absolute iodine values for masseter muscles (1.06 ± 0.4 vs. 0.96 ± 0.4 mg/mL, *p* = 0.003), the tongue (1.52 ± 0.5 vs. 1.41 ± 0.5 mg/mL, *p* = 0.04), and palatine tonsils (1.74 ± 0.7 vs. 1.57 ± 0.6 mg/mL, *p* = 0.02) were higher in women than men.

Regarding iodine ratios, women and men differed in the values for the jugular vein (higher values for women), cerebellum (higher values for men), uvula (higher values for male patients), and sublingual glands (higher ratio for men) (*p* < 0.05).

A detailed listing of all differences regarding sex is presented in Table 3 and Figure 4.

### 3.4. Impact of Age

Comparing all three age groups, the greatest differences in absolute values were observed between the youngest and oldest groups. For the carotid artery (5.48 ± 2.0 vs. 6.39 ± 2.1 mg/mL, *p* < 0.001), jugular vein (5.75 ± 1.8 vs. 7.48 ± 3.2 mg/mL, *p* < 0.001), the cerebellum (1.36 ± 0.4 vs. 1.45 ± 0.3 mg/mL, *p* = 0.02), and thyroid gland (4.10 ± 1.2 vs. 5.18 ± 1.7 mg/mL, *p* < 0.001) older patient showed higher values than the young group, while being lower for the masseter muscle (1.06 ± 0.4 vs. 0.91 ± 0.4 mg/mL, *p* < 0.001), palatine tonsils (1.74 ± 0.7 vs. 1.57 ± 0.6 mg/mL, *p* = 0.03), and lingual tonsil (1.24 ± 0.4 vs. 1.13 ± 0.5 mg/mL, *p* = 0.02). Fewer significant differences can be reported between the young and middle-aged groups (carotid artery, jugular vein, cerebellum, palatine tonsils, vocal cords, and thyroid gland) (*p* ≤ 0.03). The closest groups were the middle-aged and old age groups, with the only changes occurring in iodine uptake for the masseter muscle (*p* = 0.009), thyroid gland (*p* = 0.01), and the cervical vesicles (*p* ≤ 0.02). However, when normalized iodine ratios were analyzed, the oldest group showed the most frequently significant differences and lowest iodine ratios compared to the other age groups (*p* ≤ 0.03). A detailed demonstration of all data can be found in Table 4 and Figure 5.

### 3.5. Impact of BMI

The three subgroups defined using BMI showed the least differences in absolute and normalized data (Table 5 and Figure 6).

Absolute iodine concentrations in patients with a normal BMI were higher for the jugular vein (*p* < 0.001), parotid glands (*p* < 0.001), sublingual glands (*p* ≤ 0.02), and the sternocleidomastoid muscle (*p* < 0.001) when compared to the overweight and obese group. Additionally, the obese group had lower values for palatine tonsils, lingual tonsils, and the thyroid when compared to the group with a BMI lower than 25 (*p* ≤ 0.02). Comparing overweight patients and obese patients, the parotid gland and the lingual tonsils showed a decline with a higher BMI (*p* < 0.004).

Even less differences were documented in the iodine ratios with the jugular vein ratio in obese patients being higher than in normal-weighted patients (*p* = 0.02), parotid glands showing higher iodine values for normal-weighted patients (vs. overweight and obese) (*p* ≤ 0.003), and the sternocleidomastoid muscle having a lower iodine perfusion for overweight and obese patients, when compared to normal-weighted patients (*p* ≤ 0.004).

## 4. Discussion

In this pioneering study, we investigated the iodine concentration of cervical organs and anatomical structures to define physiologic reference values for the iodine uptake of healthy head and neck organs and tissues in a big data cohort. As indicated using DECT iodine maps, we measured absolute iodine concentrations and normalized iodine ratios for 16 structures in head and neck CT scans and identified differences related to age, gender, and weight. We discovered a broad range of iodine concentrations across the examined structures from 0.1 mg/mL to 25.1 mg/mL (13.2 mg/mL excluding the cervical vessels), with the thyroid gland exhibiting the highest uptake. Significant differences were noted regarding the iodine distribution of cervical organs concerning sex, age, and BMI.

All cervical glands showed higher absolute uptake in the female body than in men. Also, both investigated muscles showed a higher iodine density in women compared to men. Even though the amount of applied contrast agent was proportioned according to body habitus, a lower amount of contrast agent could potentially be sufficient in women when integrating iodine maps in clinical routine. This could be beneficial in reducing the risk of contrast-induced nephropathy, especially in patients with pre-existing kidney diseases, and potentially reducing the risk/severity of allergic and anaphylactic reactions to intravascular iodinated contrast media in young females, who have a higher allergic risk compared to males of the same age [12,13,14].

We observed a notable decrease in iodine uptake with age in the masseter muscle and palatine, contrasting with increased thyroid perfusion, suggesting age-related changes in iodine metabolism. A weaker and slower circulation could be the reason for the continuous increase in iodine concentrations in the vessels among young, middle-aged, and elderly groups.

Although a higher BMI results in lower iodine values in the parotid gland, there were only limited effects of weight, mainly between the normal-weighted and obese groups (significant differences regarding the sublingual gland, palatine tonsil, lingual tonsil, thyroid gland, and sternocleidomastoid muscle). We hypothesized that body habitus affects the iodine concentration in various organs and tissues due to differences in tissue composition, metabolic activity, and vascularity. Adipose tissue has a lower blood supply than lean tissue and this could impact the delivery and uptake of iodine.

The variability in iodine uptake highlights DECT’s role in enabling more personalized diagnostic and treatment strategies, aligning with the goals of precision medicine. By establishing reference values for iodine uptake, clinicians can potentially personalize diagnostic thresholds based on patient-specific factors such as age, gender, and BMI, enhancing the accuracy of diagnoses. This approach could lead to more effective treatment plans, optimized for individual patient characteristics, thereby improving outcomes.

One fundamental challenge in head and neck imaging for radiologists consists of prompt and accurate detection and interpretation of pathologies. In the last decade, approaches in DECT technology have significantly improved the visualization of tissue alterations and pathologic processes by utilizing material decomposition algorithms to investigate the atomic composition of the tissue and calculate iodine maps. Aside from morphologic, size, and shape changes, iodine quantification has enabled an additional targeted analysis of tissue characteristics and organ perfusion in contrast-enhanced CT images [2]. Lam et al. [15] reported on an improved delineation of tumor edges of squamous cell carcinoma (one of the most common malignancies of the head and neck) on iodine overlay maps [16]. In another study on squamous cell carcinoma, Kuno et al. significantly increased the specificity of CT in the differentiation between healthy and tumor-infiltrated laryngeal cartilage via the application of iodine maps. Furthermore, the quantification of iodine concentration introduced a new, highly accurate, and reproducible parameter besides Hounsfield units, the latter of which are dependent on photon energy levels, mass density, and the attenuation coefficient [3,6,17,18,19]. The application of DECT and iodine quantification in detecting and characterizing head and neck pathologies, as evidenced by the improved delineation of tumor edges and differentiation of healthy from tumor-infiltrated tissue, represents a significant advancement in diagnostic radiology. Continued development of DECT applications could lead to the establishment of new diagnostic markers and tools, facilitating early detection and treatment of malignancies.

Even though many approaches in abdominal dual-energy CT have already demonstrated that iodine quantification can determine diagnostic thresholds for benign/malignant characterization or various disease states, the application in the head and neck field is still very limited [20,21,22,23]. The up-to-date literature on DECT application in head and neck imaging is sparse, mostly focusing on categorizing different malignant cervical lymphadenopathies [24]. A study from Sauters et al.—one of the few studies where healthy structures of the head and neck area were analyzed—investigated healthy lymph nodes in the body, including cervical lymph nodes [25]. Their results (*n* = 297) of 2.09 ± 0.44 mg/mL are consistent with our definition for healthy cervical lymph nodes (1.72 ± 1.0 mg/mL). Our study, furthermore, showed that the cervical glands have a high individual variability in their iodine uptake, which may be due to several factors, for example individual glandular sizes and shapes, glandular function, and iodine uptake kinetics. This underscores the need for further investigations that facilitate a more reliable clinical application of this quantification technique in the head and neck area. In this context, our aim was to enable a more reliable clinical application of this quantification technique and, ultimately, potential implementation in future guidelines.

In our study, we observed significant variability in iodine uptake values among the normal population, underscoring the complexities of defining abnormal uptake thresholds. This variability presents challenges in distinguishing between normal physiological variations and potential pathological conditions. We propose that abnormal iodine uptake could be defined as values exceeding the mean + 2 standard deviations (SD) in a demographically matched population, a method commonly used in clinical practice for outlier detection. This approach, while statistically robust, requires careful consideration of individual patient factors such as age, gender, BMI, and overall health status. We acknowledge that this criterion, inspired by traditional clinical thresholds, may not capture all nuances of iodine metabolism but serves as a preliminary guideline. Further research is needed to validate these thresholds and to refine diagnostic criteria, considering the wide range of normal variability and the specific context of iodine-related disorders. This study highlights the importance of a nuanced approach to interpreting iodine uptake values, advocating for a balance between statistical guidelines and clinical judgment. Based on the provided data of this study, the interpretation of iodine uptake measurements and detection of pathologies in the head and neck region may be potentially optimized.

There are limitations of this study that need to be addressed. Due to biodynamical and size changes with age, not every structure could be included, e.g., pharyngeal tonsils being prominent in younger patient groups and often not measurable in older groups.

We—to the best of our knowledge—excluded all patients with known diseases or pathologies. However, there could be pre-existing or specific patient characteristics (like muscle mass, previous meals, or muscular activity and other factors) which might not have been taken into account in this study. For instance, conditions such as hypertension and diabetes can affect the delivery and distribution of contrast media, due to alterations in microvasculature and blood flow patterns. This can lead to alterations in iodine uptake in tissues, but conclusive evidence regarding the impact of these morbidities on DECT iodine uptake is currently lacking.

We would like to clarify that the main focus of our study is on iodine uptake in the cervical region. Brain structures were included in the analysis because they were in the scan field and constituted part of the imaging datasets. We acknowledge that the inherent nonhomogeneity of the brain structures, due to gray and white matter as well as location, and the presence of the blood–brain barrier may affect iodine uptake.

CT scans, as well as post-processing, were performed using one manufacturer (Siemens Healthineers), restricting the generalization of our results to other systems. Differences may also occur when different scan and contrast media application protocols are chosen. It should be noted that the presented study was applied to portal venous CT datasets; using different scan phases will cause different outcomes.

In this study, a predominantly Caucasian study population was included; there is a need for more diverse studies to establish more comprehensive referencing. Because no statistical error propagation was performed, the results could have been affected by calculation errors.

## 5. Conclusion

In conclusion, this study provides reference values of iodine concentrations in healthy head and neck structures, valid for examinations with a delay of 70 s after injection. The scatter and the differences between sex, age groups, and BMI should be considered when performing iodine measurements. Particularly, care should be taken during the interpretation of iodine values in clinical settings.

## Figures and Tables

**Figure 1 diagnostics-14-00496-f001:**
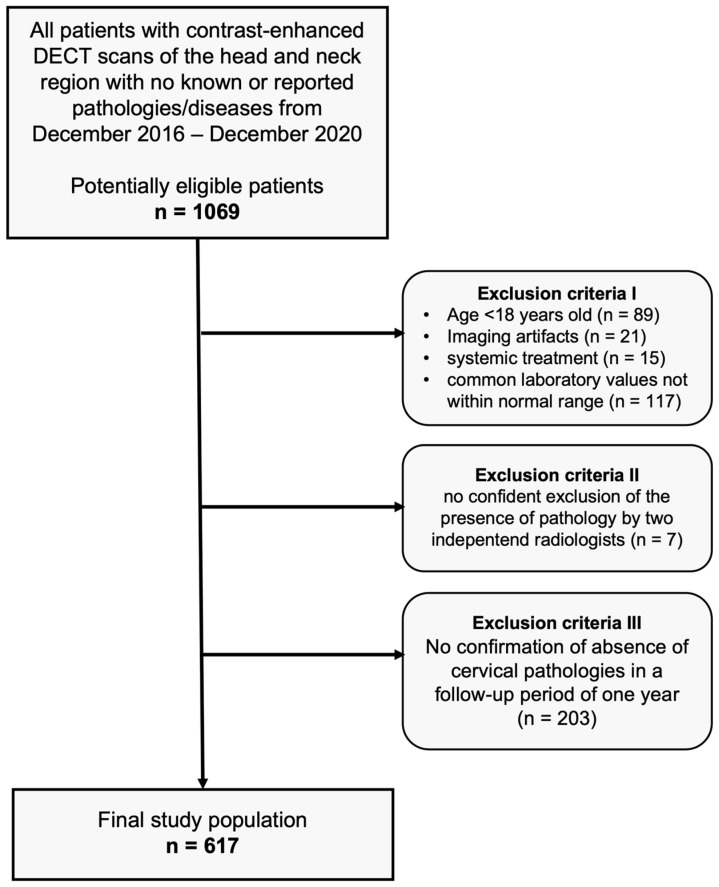
Flowchart of study inclusion.

**Figure 2 diagnostics-14-00496-f002:**
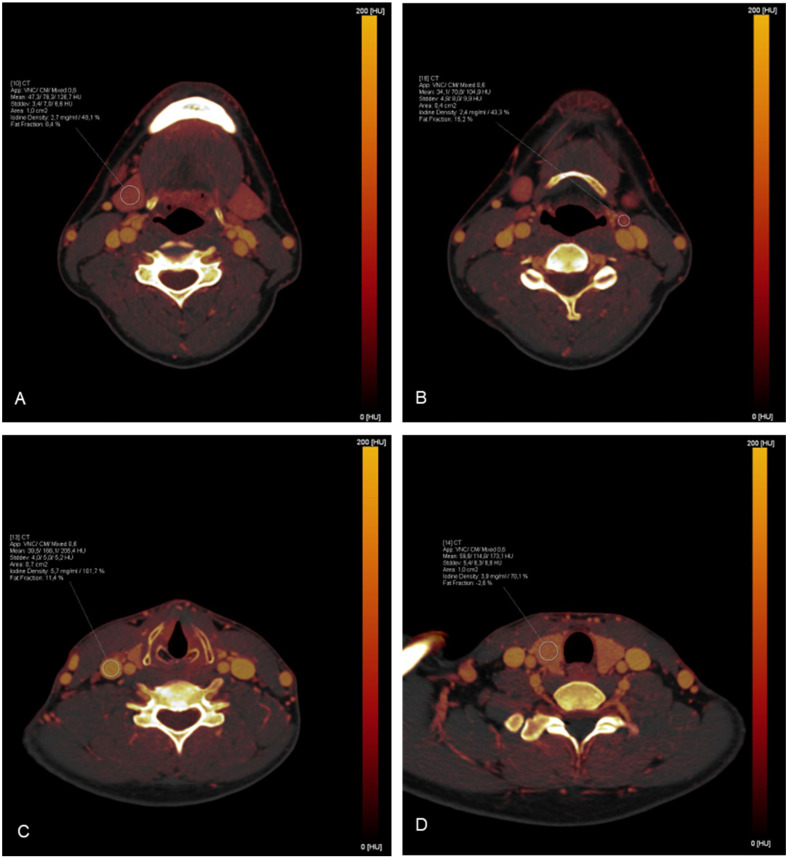
Exemplary demonstration of ROI measurements on iodine maps: (**A**) The right submandibular gland; (**B**) A lymph node in cervical level II on the left side; (**C**) The right jugular vein; (**D**) The right lobe of the thyroid gland.

**Figure 3 diagnostics-14-00496-f003:**
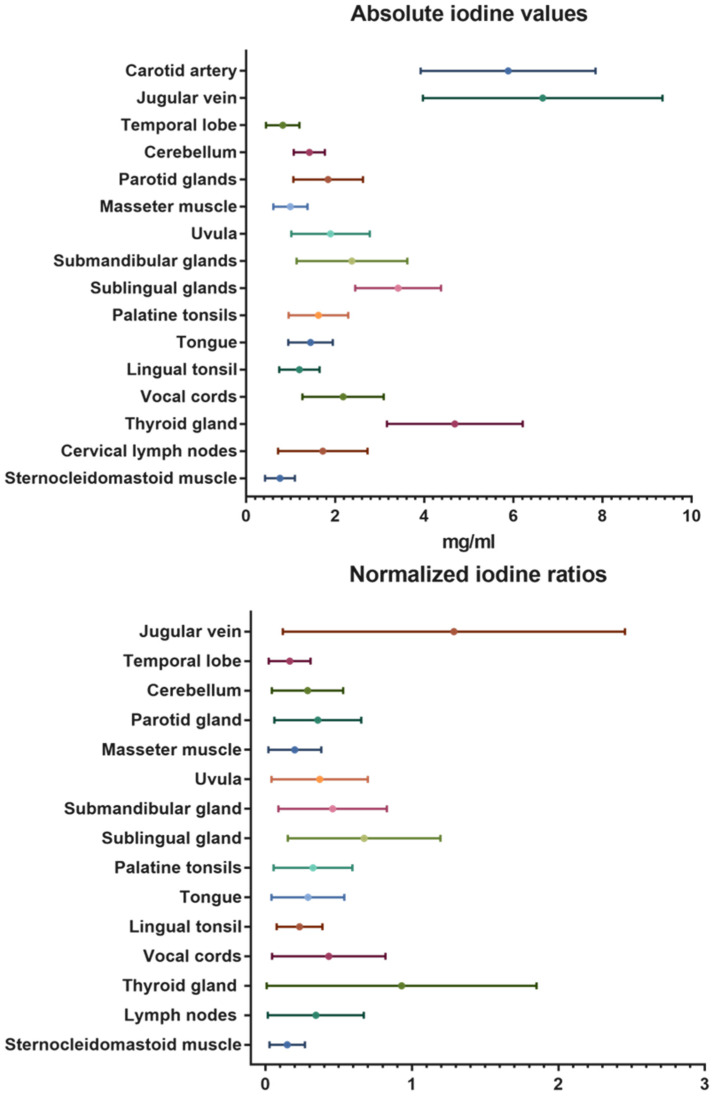
Absolute iodine values and normalized iodine ratios for the different organs and tissues of the head and neck.

**Figure 4 diagnostics-14-00496-f004:**
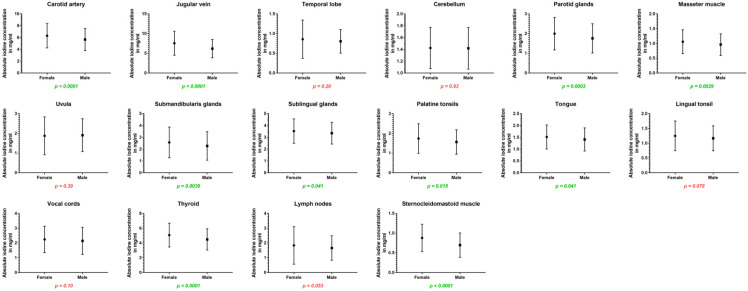
Column bars (mean with SD) of absolute iodine values for female vs. male with *p*-values (green indicating significant differences).

**Figure 5 diagnostics-14-00496-f005:**
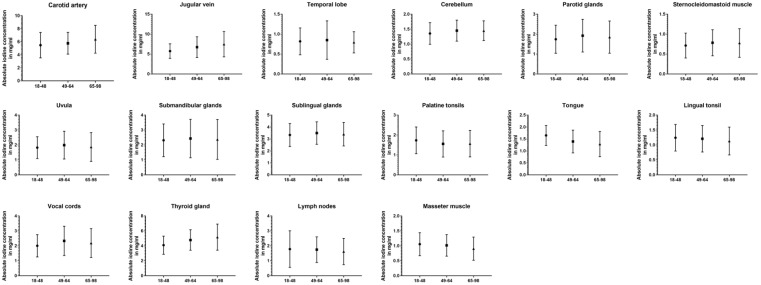
Column bars (mean with SD) of absolute iodine values for young vs. middle-aged vs. old subgroups.

**Figure 6 diagnostics-14-00496-f006:**
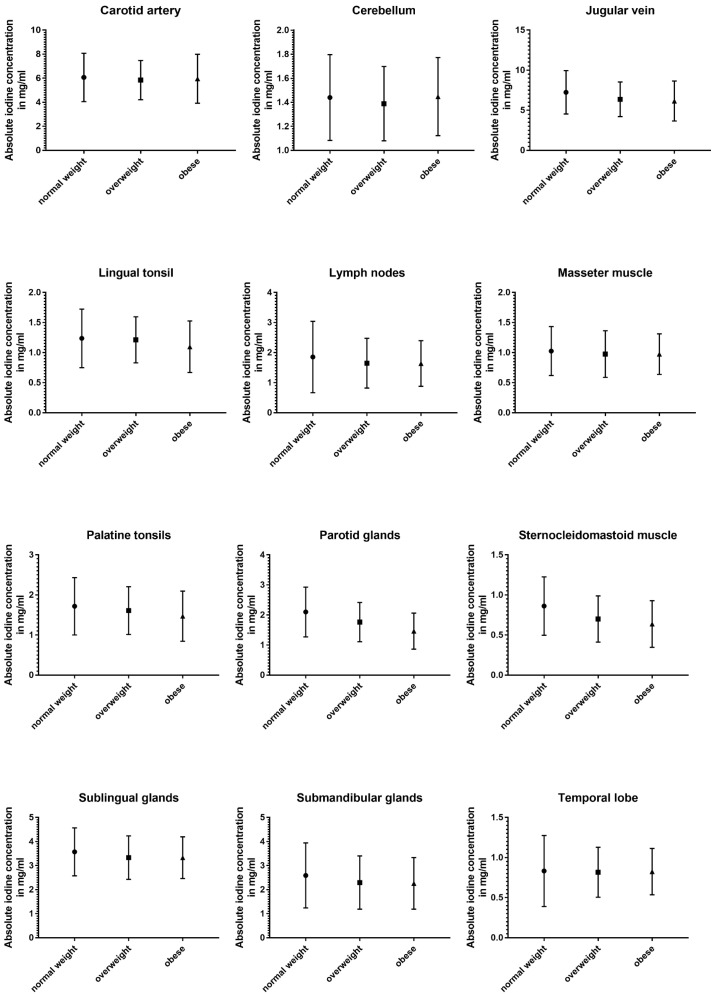
Column bars (mean with SD) of absolute iodine uptake for normal weight vs. overweight vs. obese subgroups.

**Table 1 diagnostics-14-00496-t001:** Listing of the 35 ROI measurements with a corresponding description of their placements.

Structure	Number of ROI	ROI Placement
Carotid artery	2	central within the vessel at the level of the thyroid gland
Jugular vein	2	central within the vessel at the level of the thyroid gland
Temporal lobe	2	right and left temporal lobe
Cerebellum	2	right and left cerebellum
Parotid glands	2	right and left gland, avoiding duct
Masseter muscle	2	the superficial portion on both sides
Uvula	1	distal of the soft palatine
Submandibular glands	2	right and left gland, avoiding duct
Sublingual glands	2	right and left gland, avoiding surrounding tissue
Palatine tonsils	2	at the most prominent portion
Tongue	2	right and left half of the tongue
Lingual tonsil	1	at the most prominent portion
Vocal cords	2	at the level of arytenoid cartilage
Thyroid gland	4	upper and lower portion, both sides
Cervical lymph nodes	5	one on each cervical lymph node level
Sternocleidomastoid muscle	2	common muscle belly

**Table 2 diagnostics-14-00496-t002:** Absolute iodine concentration and normalized iodine ratio values for the different structures of the head and neck.

Structure	Absolute Iodine Concentration in mg/mL (Mean ± SD)	Normalized Iodine Ratio
Carotid artery	5.88 ± 2.0	1
Jugular vein	6.66 ± 2.7	1.29 ± 1.2
Temporal lobe	0.82 ± 0.4	0.17 ± 0.1
Cerebellum	1.41 ± 0.4	0.29 ± 0.2
Parotid glands	1.84 ± 0.8	0.36 ± 0.3
Masseter muscle	0.99 ± 0.4	0.21 ± 0.2
Uvula	1.90 ± 0.9	0.37 ± 0.3
Submandibular glands	2.37 ± 1.2	0.46 ± 0.4
Sublingual glands	3.41 ± 1.0	0.67 ± 0.5
Palatine tonsils	1.62 ± 0.7	0.33 ± 0.3
Tongue	1.45 ± 0.5	0.29 ± 0.2
Lingual tonsil	1.20 ± 0.5	0.23 ± 0.2
Vocal cords	2.18 ± 0.9	0.43 ± 0.4
Thyroid gland	4.68 ± 1.5	0.93 ± 0.9
Cervical lymph nodes	1.72 ± 1.0	0.35 ± 0.3
Sternocleidomastoid muscle	0.76 ± 0.3	0.15 ± 0.1

**Table 3 diagnostics-14-00496-t003:** Absolute iodine concentration and normalized iodine ratio values for female vs. male (asterisk indicates statistical significance).

Structure	Absolute Iodine Concentration in mg/mL (Mean ± SD)		*p*-Values
	Female	Male	
Carotid artery	6.31 ± 2.1	5.66 ± 1.8	<0.001 *
Jugular vein	7.56 ± 3.1	6.18 ± 2.3	<0.001 *
Temporal lobe	0.86 ± 0.5	0.80 ± 0.3	0.199
Cerebellum	1.42 ± 0.3	1.42 ± 0.4	0.926
Parotid glands	2.00 ± 0.8	1.76 ± 0.7	<0.001 *
Masseter muscle	1.06 ± 0.4	0.96 ± 0.4	0.003 *
Uvula	1.88 ± 1.0	1.91 ± 0.8	0.387
Submandibular glands	2.57 ± 1.3	2.27 ± 1.2	0.004 *
Sublingual glands	3.53 ± 1.0	3.35 ± 0.9	0.041 *
Palatine tonsils	1.74 ± 0.7	1.57 ± 0.6	0.018 *
Tongue	1.52 ± 0.5	1.41 ± 0.5	0.041 *
Lingual tonsil	1.25 ± 0.5	1.17 ± 0.4	0.078
Vocal cords	2.24 ± 0.9	2.14 ± 0.9	0.100
Thyroid gland	5.07 ± 1.6	4.48 ± 1.4	<0.001 *
Cervical lymph nodes	1.83 ± 1.3	1.66 ± 0.8	0.053
Sternocleidomastoid muscle	0.87 ± 0.3	0.70 ± 0.3	<0.001 *
Structure	Normalized Iodine Ratios		*p*-Values
	Female	Male	
Jugular vein	1.38 ± 1.3	1.24 ± 1.1	0.008 *
Temporal lobe	0.17 ± 0.2	0.17 ± 0.1	0.073
Cerebellum	0.27 ± 0.2	0.30 ± 0.3	<0.001 *
Parotid glands	0.36 ± 0.2	0.37 ± 0.4	0.54
Masseter muscle	0.20 ± 0.2	0.21 ± 0.2	0.99
Uvula	0.34 ± 0.3	0.39 ± 0.3	0.001 *
Submandibular glands	0.46 ± 0.3	0.46 ± 0.4	0.53
Sublingual glands	0.65 ± 0.4	0.71 ± 0.6	0.049 *
Palatine tonsils	0.31 ± 0.2	0.34 ± 0.3	0.76
Tongue	0.29 ± 0.3	0.29 ± 0.2	0.48
Lingual tonsil	0.23 ± 0.1	0.24 ± 0.2	0.18
Vocal cords	0.41 ± 0.3	0.45 ± 0.5	0.35
Thyroid gland	0.94 ± 0.9	0.93 ± 0.9	0.72
Cervical lymph nodes	0.34 ± 0.3	0.36 ± 0.5	0.67
Sternocleidomastoid muscle	0.16 ± 0.1	0.15 ± 0.1	0.22

**Table 4 diagnostics-14-00496-t004:** Absolute iodine concentration and normalized iodine ratio values for young vs. middle-aged vs. old subgroups (asterisk indicates statistical significance).

Structure	Absolute Iodine Concentration in mg/mL (Mean ± SD)			*p*-Values		
	18–48 Years	49–64 Years	65–98 Years	18–48 vs. 49–64	18–48 vs. 65–98	49–64 vs. 65–98
Carotid artery	5.48 ± 2.0	5.77 ± 1.7	6.39 ± 2.1	0.027 *	<0.001 *	0.008 *
Jugular vein	5.75 ± 1.8	6.73 ± 2.6	7.48 ± 3.2	<0.001 *	<0.001 *	0.024 *
Temporal lobe	0.82 ± 0.3	0.85 ± 0.5	0.80 ± 0.3	0.69	0.74	0.45
Cerebellum	1.36 ± 0.4	1.45 ± 0.4	1.45 ± 0.3	0.032 *	0.022 *	0.93
Parotid glands	1.75 ± 0.7	1.92 ± 0.8	1.85 ± 0.8	0.099	0.86	0.86
Masseter muscle	1.06 ± 0.4	1.02 ± 0.4	0.91 ± 0.4	0.53	<0.001 *	0.009 *
Uvula	1.83 ± 0.7	1.99 ± 0.9	1.87 ± 1.0	0.35	0.83	0.26
Submandibular glands	2.32 ± 1.1	2.44 ± 1.3	2.37 ± 1.3	0.39	0.93	0.54
Sublingual glands	3.34 ± 1.0	3.50 ± 0.9	3.40 ± 1.0	0.22	0.78	0.58
Palatine tonsils	1.74 ± 0.7	1.55 ± 0.7	1.57 ± 0.6	0.008 *	0.031 *	0.75
Tongue	1.65 ± 0.4	1.39 ± 0.5	1.29 ± 0.5	<0.001	<0.001	0.034
Lingual tonsil	1.24 ± 0.4	1.21 ± 0.4	1.13 ± 0.5	0.94	0.015 *	0.22
Vocal cords	2.01 ± 0.7	2.33 ± 1.0	2.19 ± 1.0	0.003 *	0.43	0.21
Thyroid gland	4.10 ± 1.2	4.79 ± 1.4	5.18 ± 1.7	<0.001 *	<0.001 *	0.01 *
Cervical lymph nodes	1.79 ± 1.2	1.75 ± 0.9	1.62 ± 0.9	0.75	0.53	0.31
Sternocleidomastoid muscle	0.71 ± 0.3	0.78 ± 0.3	0.78 ± 0.4	0.083	0.28	0.61
Structure	Normalized Iodine Ratio (Mean ± SD)			*p*-Values		
	18–48 years	49–64 years	65–98 years	18–48 vs. 49–64	18–48 vs. 65–98	49–64 vs. 65–98
Jugular vein	1.27 ± 1.4	1.27 ± 0.9	1.32 ± 1.0	0.11	0.099	0.87
Temporal lobe	0.18 ± 0.2	0.17 ± 0.2	0.15 ± 0.1	0.54	0.004 *	0.023 *
Cerebellum	0.29 ± 0.2	0.30 ± 0.3	0.29 ± 0.4	0.75	0.034 *	0.077
Parotid glands	0.35 ± 0.2	0.37 ± 0.3	0.36 ± 0.4	0.92	0.03 *	0.023 *
Masseter muscle	0.22 ± 0.2	0.21 ± 0.2	0.18 ± 0.3	0.041 *	<0.001 *	0.001 *
Uvula	0.38 ± 0.2	0.39 ± 0.4	0.34 ± 0.4	0.78	0.004 *	0.01 *
Submandibular glands	0.47 ± 0.3	0.47 ± 0.4	0.44 ± 0.4	0.52	0.038 *	0.21
Sublingual glands	0.68 ± 0.3	0.71 ± 0.7	0.66 ± 0.7	0.86	<0.001 *	0.004 *
Palatine tonsils	0.35 ± 0.2	0.32 ± 0.3	0.33 ± 0.4	<0.001 *	<0.001 *	0.83
Tongue	0.36 ± 0.3	0.28 ± 0.2	0.23 ± 0.2	<0.001 *	<0.001 *	<0.001 *
Lingual tonsil	0.26 ± 0.2	0.24 ± 0.2	0.21 ± 0.2	0.035 *	<0.001 *	0.009 *
Vocal cords	0.43 ± 0.4	0.46 ± 0.4	0.42 ± 0.4	0.95	0.085	0.004 *
Thyroid gland	0.94 ± 1.2	0.92 ± 0.6	0.95 ± 0.9	0.035 *	0.12	0.69
Cervical lymph nodes	0.37 ± 0.3	0.36 ± 0.4	0.33 ± 0.5	0.91	0.001 *	0.033 *
Sternocleidomastoid muscle	0.15 ± 0.1	0.16 ± 0.1	0.14 ± 0.1	0.65	0.25	0.082

**Table 5 diagnostics-14-00496-t005:** Absolute iodine concentration and normalized iodine ratio values for normal-weighted vs. overweight vs. obese subgroups (asterisk indicates statistical significance).

Structure	Absolute Iodine Concentration in mg/mL (Mean ± SD)			*p*-Values		
	No BMI < 25	Ov 25–29.9	Ob BMI > 29.9	No vs. Ov	No vs. Ob	Ov vs. Ob
Carotid artery	6.1 ± 2.0	5.84 ± 1.6	5.96 ± 2.0	0.91	0.59	0.84
Jugular vein	7.24 ± 2.7	6.37 ± 2.1	6.16 ± 2.5	<0.001 *	<0.001 *	0.87
Temporal lobe	0.83 ± 0.4	0.82 ± 0.3	0.83 ± 0.3	0.83	0.54	0.50
Cerebellum	1.44 ± 0.4	1.39 ± 0.3	1.45 ± 0.3	0.67	0.79	0.69
Parotid glands	2.10 ± 0.8	1.76 ± 0.7	1.46 ± 0.6	<0.001 *	<0.001 *	0.004 *
Masseter muscle	1.03 ± 0.4	0.98 ± 0.4	0.98 ± 0.3	0.22	0.60	0.79
Uvula	1.97 ± 1.0	1.91 ± 0.9	1.78 ± 0.7	0.56	0.51	0.39
Submandibular glands	2.59 ± 1.4	2.29 ± 1.1	2.26 ± 1.1	0.08	0.092	0.69
Sublingual glands	3.57 ± 1.0	3.33 ± 0.9	3.33 ± 0.9	0.014 *	0.024 *	0.87
Palatine tonsils	1.72 ± 0.7	1.61 ± 0.6	1.47 ± 0.6	0.29	0.014 *	0.29
Tongue	1.45 ± 0.5	1.48 ± 0.5	1.41 ± 0.5	0.83	0.63	0.53
Lingual tonsil	1.24 ± 0.5	1.21 ± 0.4	1.10 ± 0.4	0.58	0.006 *	0.038 *
Vocal cords	2.27 ± 0.9	2.14 ± 0.8	2.27 ± 1.2	0.40	0.40	0.80
Thyroid gland	4.99 ± 1.5	4.65 ± 1.5	4.41 ± 1.3	0.068	0.002 *	0.52
Cervical lymph nodes	1.85 ± 1.2	1.65 ± 0.8	1.64 ± 0.8	0.062	0.11	0.92
Sternocleidomastoid muscle	0.86 ± 0.4	0.70 ± 0.3	0.64 ± 0.3	<0.001 *	<0.001 *	0.64
Structure	Normalized Iodine Ratio (Mean ± SD)			*p*-Values		
	No BMI < 25	Ov 25–29.9	Ob BMI > 29.9	No vs. Ov	No vs. Ob	Ov vs. Ob
Jugular vein	1.39 ± 1.3	1.15 ± 0.6	1.26 ± 1.3	0.051	0.017 *	0.70
Temporal lobe	0.17 ± 0.2	0.16 ± 0.1	0.18 ± 0.2	0.48	0.36	0.85
Cerebellum	0.29 ± 0.3	0.27 ± 0.3	0.32 ± 0.3	0.73	0.23	0.53
Parotid glands	0.41 ± 0.4	0.34 ± 0.3	0.32 ± 0.3	0.003 *	<0.001 *	0.69
Masseter muscle	0.20 ± 0.2	0.20 ± 0.3	0.22 ± 0.2	0.51	0.65	0.87
Uvula	0.39 ± 0.4	0.35 ± 0.2	0.37 ± 0.3	0.99	0.99	0.97
Submandibular glands	0.50 ± 0.4	0.41 ± 0.2	0.45 ± 0.4	0.37	0.16	0.63
Sublingual glands	0.69 ± 0.5	0.64 ± 0.6	0.72 ± 0.7	0.69	0.99	0.92
Palatine tonsils	0.34 ± 0.3	0.32 ± 0.4	0.33 ± 0.3	0.49	0.33	0.31
Tongue	0.29 ± 0.3	0.27 ± 0.1	0.31 ± 0.3	0.40	0.37	0.61
Lingual tonsil	0.23 ± 0.2	0.23 ± 0.1	0.24 ± 0.2	0.87	0.52	0.40
Vocal cords	0.44 ± 0.4	0.40 ± 0.3	0.46 ± 0.5	0.41	0.61	0.98
Thyroid gland	0.98 ± 0.9	0.87 ± 0.5	0.94 ± 0.9	0.43	0.71	0.74
Cervical lymph nodes	0.36 ± 0.4	0.33 ± 0.5	0.36 ± 0.4	0.34	0.75	0.75
Sternocleidomastoid muscle	0.17 ± 0.1	0.14 ± 0.1	0.14 ± 0.1	0.003 *	0.004 *	0.92

## Data Availability

The data presented in this study are available on request from the corresponding author. The data are not publicly available due to data protection.

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
