# Peer review of "Dual-Energy CT Iodine Uptake of Head and Neck: Definition of Reference Values in a Big Data Cohort"

_diagnostics, 2024, doi:10.3390/diagnostics14050496_

Round 1

Reviewer 1 Report

Comments and Suggestions for Authors

Specific comments:

1. Figure 4 and 5 are truncated on the right in the pdf file that is available to me.

2. From figures 4 to 6 and the tables in the results, the variability of the iodine uptake values are wide among the normal population. Can the authors comment on how one should use these values as reference for detecting abnormal uptake? For example, are abnormal uptake expected to be above the mean + 2 stdev in demographically matched population?

Reviewer 2 Report

Comments and Suggestions for Authors

This study aimed to define the physiologic iodine uptake values in the head and neck region using dual-energy-CT (DECT) scans, addressing a gap in the literature regarding reference values for healthy organs. The results show that iodine uptake varies significantly across different organs and tissues and is influenced by gender, age, and BMI. Key findings include higher iodine concentrations in women in specific areas, increased uptake with age in certain organs, and variations in iodine levels based on BMI. These variations underline the importance of considering gender, age, and BMI in the clinical interpretation of head and neck DECT scans.

Dear Author, 
Overall, the study is well documented. I recommend expanding the introduction, including more references to present a sufficient background for the reader.

Please deepen the discussion on the clinical implications of your findings. Additionally, improve the presentation of your figures. Finally, consider comparing your findings with existing literature to highlight the study's contribution to the field.

Thank you

Comments on the Quality of English Language

Minor editing of English language required
